# Relationship Between Periodontal Disease and Systemic Diseases in Non-Human Primates

**DOI:** 10.3390/vetsci12080784

**Published:** 2025-08-21

**Authors:** Bruno Pires Miranda, Amanda Figueira da Silva, Júlia de Castro Ascenção, Rhagner Bonono dos Reis, Marcio Vinícius Marins Teixeira, Marcos Tobias de Santana Miglionico, Helena Lúcia Carneiro Santos

**Affiliations:** Oswaldo Cruz Institute, Rio de Janeiro 21040-360, RJ, Brazil; amanda.figueirabio@gmail.com (A.F.d.S.); julia_castro@id.uff.br (J.d.C.A.); bonono.rhagner@gmail.com (R.B.d.R.); vinicius_marinsteixeira@hotmail.com (M.V.M.T.); miglioni@gmail.com (M.T.d.S.M.); helenalucias@ioc.fiocruz.br (H.L.C.S.)

**Keywords:** inflammatory response, microbiota, oral cavity

## Abstract

Periodontal disease in non-human primates (NHPs) has been associated with the emergence and/or worsening of several systemic diseases, but there are still few studies investigating this relationship. This study aims to analyze the relationship between periodontal disease and the emergence of systemic diseases in NHPs, seeking to understand the mechanisms involved and the possible implications for their health and clinical management. For this purpose, an integrative review of the literature was carried out, using the PICO strategy for the analysis of the articles. Clinical trials, cohort studies, observational studies, case reports and systematic and integrative reviews were included, while narrative reviews were excluded. The results indicated a significant relationship between periodontal disease and several systemic conditions, such as diabetes, cardiovascular diseases, arthritis and gestational complications. The literature also suggested that the inflammatory mechanisms resulting from periodontal disease may contribute to the worsening of these conditions. It is concluded that periodontal disease in NHPs has important implications for systemic health and that continued investigation of the underlying mechanisms is essential for the development of more effective clinical management strategies.

## 1. Introduction

Currently, periodontal disease is considered by international guidelines to be a chronic noncommunicable inflammatory condition (NCD) characterized by the destruction of tooth-supporting tissues, including the periodontium and alveolar bone, in addition to the formation of periodontal pockets and bleeding on probing [1], which is directly associated with the oral microbiome. In addition to its local impact, this condition has been widely related to several systemic diseases [2,3]. After all, oral health plays a fundamental role in maintaining the systemic balance of the body. In recent years, the association between periodontal disease and the development of several non-infectious systemic diseases has received increasing attention in the scientific literature [3,4,5].

Periodontal disease has long been recognized as a chronic inflammatory condition that affects the supporting structures of the teeth, leading to clinical consequences such as gingival bleeding, periodontal pocket formation, tooth mobility, and eventual tooth loss. These local manifestations can significantly impair oral function, quality of life, and overall well-being. In addition to these well-established local effects, a substantial body of research developed over the past several decades has demonstrated the systemic implications of periodontal disease. It is now widely accepted that periodontal disease can contribute to or exacerbate various NCDs, such as adverse pregnancy outcomes, cardiovascular disease, and diabetes. Dental and medical professionals, as well as researchers, are encouraged to continue expanding their perspective on periodontal disease, not only as a localized condition but also as one that plays a role in systemic health, either as a comorbidity or as a contributing factor to the development and progression of other diseases [6].

The systemic expression of periodontal disease is driven by a combination of bacterial dissemination and chronic inflammation. Periodontal damage, particularly with the formation of deep periodontal pockets, creates an anaerobic microenvironment that promotes the proliferation of pathogenic microorganisms and the persistent activation of the host immune response. This leads to the local release of a variety of pro-inflammatory mediators and enzymes, such as C-reactive protein (CRP), matrix metalloproteinases (MMPs), interleukins (IL-1β, IL-6, IL-10), and tumor necrosis factor-alpha (TNF-α). These factors, along with bacterial products, are responsible for the destruction of soft tissues such as gingiva, leading to gingival recession and the consequent exposure of tooth roots [7].

Additionally, bacterial components, particularly lipopolysaccharides (LPS) from the outer membranes of Gram-negative bacteria, stimulate the host immune response and directly activate osteoclasts, promoting bone resorption (osteolysis) and contributing to the progressive loss of alveolar bone support. As these mediators and bacterial by-products accumulate, they not only exacerbate local tissue destruction but may also spill over into the systemic circulation, giving rise to a chronic low-grade inflammatory state. In some cases, periodontal pathogens may enter the bloodstream through ulcerated epithelium, resulting in transient bacteremia and the potential for bacterial embolization to distant organs [8]. The combined effect of microbial dissemination and systemic inflammation, sometimes referred to as a “cytokine storm”, has been implicated in the pathophysiology of various noncommunicable diseases. This understanding gave rise to the concept of “periodontal medicine,” supported by numerous epidemiological, experimental, and interventional studies that demonstrate how periodontal disease may negatively influence systemic health [7].

Evidence from human interventional studies reinforces this possible cause-and-effect relationship, demonstrating that local treatment of periodontal disease can lead to a reduction in systemic inflammation and a decrease in biomarkers associated with concomitant diseases. Furthermore, research conducted in preclinical models has provided important mechanistic support for this connection, suggesting that periodontal disease can directly contribute to the progression of several systemic pathologies, including autoimmune, cardiometabolic, cognitive and neurodegenerative disorders; respiratory infections; and even certain types of cancer. Epidemiological studies point to a strong correlation between periodontal disease and these diseases, raising the hypothesis that chronic inflammation triggered by periodontal infection may act as a causal factor in the development of systemic comorbidities [2,5,9,10].

According to the World Organization of National Colleges, Academies and Academic Associations of General Practitioners/Family Physicians (WONCA), periodontal disease is independently associated with several systemic conditions, such as cardiovascular disease, chronic obstructive pulmonary disease, diabetes, obstructive sleep apnea and complications of COVID-19 [1]. It has also been shown that changes in the oral microbiome can serve as potential biomarkers to predict the development of systemic diseases that affect bones [11]. On the other hand, its treatment has been related to improvements in systemic health outcomes [10,12].

As in humans, it is essential to raise awareness of periodontal diseases in animals, including their implications and associated risk factors [1]. It should be noted that there is a connection between periodontal disease and systemic diseases. This connectivity can be attributed to the spread of inflammation, microorganisms and microbial products to distant organs. Oral bacteria can reach the intestine through swallowed saliva, inducing intestinal dysbiosis and gastrointestinal dysfunctions. Periodontal pathogens such as *Aggregatibacter actinomycetemcomitans*, *Fusobacterium nucleatum*, *Haemophilus*, *Helicobacter pylori, Klebsiella*, *Parvimonas micra*, *Porphyromonas gingivalis*, *Peptostreptococcus*, *Streptococcus mutans* and *Veillonella* can resist the acidic environment, survive in the intestine and cause intestinal dysbiosis. Intestinal dysbiosis, in turn, increases inflammation in the gastrointestinal tract and induces dysplastic changes that result in intestinal dysfunction. Studies have already associated oral bacteria and the oral–intestinal axis with several disorders of the gastrointestinal system, such as inflammatory bowel diseases, liver diseases, hepatocellular and pancreatic ductal carcinoma, ulcerative colitis and Crohn’s disease [7,8,13,14].

Among the systemic consequences of periodontal disease, the association with the development of neoplasms in distinct anatomical sites, such as the oral cavity, lungs and colon, has become an area of growing scientific interest. This relationship is multifactorial and involves more complex and diverse pathophysiological mechanisms compared to other systemic outcomes. Chronic inflammation related to periodontal disease can foster a pro-tumorigenic microenvironment by inducing oxidative stress, promoting genetic mutations, and altering cellular signaling pathways involved in cell proliferation and apoptosis. Furthermore, specific periodontal pathogens, such as *Fusobacterium nucleatum*, have been detected in tumor tissues, particularly in cases of colorectal cancer, suggesting a possible microbial contribution to carcinogenesis. Due to the intricate interplay between local infection, systemic inflammation and tumor biology, this association warrants special attention and further investigation within the field of periodontal medicine [14,15,16].

The relationship with cardiovascular diseases has also been widely investigated through observational studies, systematic reviews and meta-analyses [4,17]. Evidence indicates a significant link between periodontal disease and conditions such as atrial fibrillation, cerebrovascular disease, heart failure, ischemic heart disease and peripheral arterial disease. The main mechanism proposed for this association involves systemic inflammation triggered by periodontal infection. The chronic inflammatory response creates an environment conducive to the development of atherosclerosis, through the activation of inflammatory markers, damage mediated by antibodies that cross-react with components of the vascular wall and the direct invasion of periodontal pathogens into the bloodstream. Furthermore, studies demonstrate a correlation between the severity of periodontal disease and the increase in inflammatory biomarkers, reinforcing the hypothesis that this condition may contribute to the worsening of cardiovascular diseases [4].

The possible causal relationship between periodontal disease and its comorbidities has been corroborated by recent animal studies, which identify biologically plausible and clinically consistent mechanisms by which this condition can trigger or aggravate systemic diseases. Understanding how certain extraoral pathologies are influenced by the dissemination of periodontal pathogens and systemic inflammation associated with periodontal disease may pave the way for new therapeutic approaches aimed at reducing the risk of related comorbidities, in both humans and animals [9].

Due to their close similarity to humans, non-human primates (NHPs) are widely used as research models in studies related to human health. In this context, there is a substantial body of research on periodontal disease in these animals, as they develop the condition with clinical features and systemic consequences comparable to those observed in humans [18].

The literature has emphasized that inadequate or absent oral hygiene is a key predisposing factor in the development of periodontal disease. Dental biofilm, composed primarily of bacteria, begins to mature within 24 h if not removed. This maturation process results in the organization of a structured and pathogenic biofilm that adheres to the tooth surface and gingival margin. If regular mechanical disruption—such as tooth brushing—is not performed, the bacterial biofilm becomes more complex and virulent. This persistent presence of plaque triggers the host’s immune response, initiating an inflammatory cascade aimed at controlling the infection, which, if unresolved, can lead to the progressive destruction of periodontal tissues.

However, in the NHP population, some animals are susceptible to the disease, while others do not develop it. The absence of the disease in a group of NHPs from the same enclosure indicates that oral hygiene may be a factor in accelerating the course of the disease, but it is not the main cause of the development of the disease. Given this evidence, it is essential to understand the relationship between periodontal disease and systemic health in NHPs. In this context, this study aims to analyze the relationship between periodontal disease and the emergence of systemic diseases in these animals, seeking to understand the mechanisms involved and the possible implications for their health and clinical management, based on available scientific evidence.

## 2. Materials and Methods

This study was prepared through an integrative literature review, using the PICO strategy (Population, Intervention, Comparison and Outcome) as a criterion for analysis and inclusion of articles, defined as follows:−Population (P): Non-human primates with periodontal disease.−Intervention (I): Diagnosis, management or treatment of periodontal disease.−Comparison (C): Primates without periodontal disease.−Outcome (O): Occurrence and progression of systemic diseases associated with periodontal disease.

Clinical trials, cohort studies, observational studies, case reports and systematic and integrative reviews were included. Narrative literature reviews were excluded. The research used combinations of the following descriptors for the search: periodontal disease; periodontitis; systemic diseases; primate; marmoset; baboon; monkeys; chimpanzees; bonobos; gorillas; orangutans.

To ensure a comprehensive search, the literature review was conducted using well-established scientific databases, including PubMed, SciELO and the Virtual Health Library (BVS), as well as Google Scholar. All potentially relevant articles available in these platforms were considered, without restrictions on the year of publication. The initial screening was based on the reading of titles and abstracts to identify studies aligned with the objective of this review. Articles were included if they addressed the relationship between periodontal disease and systemic manifestations in non-human primates, according to the defined PICO strategy. Studies that presented relevant information were selected for full-text reading and inclusion, resulting in a total of 11 articles that met all eligibility criteria and contributed effectively to the analysis proposed by this review.

## 3. Results

A total of 11 articles were selected, the oldest being published in 1980 and the most recent in 2022. These articles are presented in Table 1, according to their year of publication, authors, country of research, objective, method, species analyzed and associated systemic disease(s) identified.

It can be observed that among the eleven articles, the vast majority (7; 64%) were conducted in the United States, with two others in China (18%), one in Brazil (9%) and one in the United Kingdom (9%). There were four (36%) experimental articles, one integrative review (9%) and six (55%) cross-sectional observational studies. All of them demonstrated relationships between periodontal disease and other systemic conditions. The conclusions of these studies are discussed below.

## 4. Discussion

Pettigrew et al.’s [19] findings (demonstrated radiographic quantification methods) provide relevant evidence for understanding the relationship between periodontal disease and the emergence of systemic pathologies in NHPs. Their findings indicate that diabetic rhesus monkeys present greater bone loss and widening of the periodontal ligament, with a positive correlation between hyperglycemia and bone resorption. These results suggest that periodontal disease may act as an aggravating factor in the systemic impairment of these animals, possibly reducing their resistance to infections and contributing to the progression of metabolic conditions, such as diabetes. Still in relation to the aforementioned study, the identification of a quantitative system for measuring these radiographic changes reinforces the importance of objective methods in the assessment of periodontal health and its interconnection with systemic diseases. Thus, the researchers corroborate the need to investigate the underlying mechanisms of this relationship, providing support for understanding the clinical implications and management of oral and systemic health in NHPs.

Aufdemorte et al. [20] demonstrated that osteoporosis and oral bone loss are interrelated, especially in aging populations, which may have important implications for oral health and prosthetic rehabilitation in elderly patients. The study demonstrated that bone mineral density in the jaws of aged baboons was significantly reduced compared with young controls, and was also correlated with ovarian dysfunction, a factor known to affect systemic bone health. The authors compare this with data from human studies, in which progressive alveolar bone loss, observed in elderly women with osteoporosis, reflects these systemic changes and may further complicate dental rehabilitation, especially in cases of edentulism, where maxillary and mandibular bone atrophy prevents the installation of adequate prostheses. These data suggest that the investigation of the relationships between systemic osteoporosis and oral bone loss related to periodontal disease should be further investigated, as they may provide important information for more effective and personalized treatments in geriatric dentistry.

Another important finding from previous studies is that periodontal disease can contribute to systemic inflammatory and metabolic processes, reinforcing its association with chronic diseases. The study by Ebersole et al. [21] in NHPs demonstrated that periodontal disease is associated with elevated serum endotoxins (lipopolysaccharides); increased systemic inflammatory biomarkers, including C-reactive protein, fibrinogen and interleukin (IL)-8; and significant alterations in lipid metabolism. These lipid changes include increased levels of total cholesterol, triglycerides and low-density lipoprotein (LDL), along with reduced levels of high-density lipoprotein (HDL), characterizing a state of dyslipidemia. Such findings support the hypothesis that chronic periodontal inflammation may act as a contributing risk factor for the development or worsening of systemic conditions, particularly those related to cardiovascular and metabolic health. These results emphasize the importance of an integrative perspective on periodontal health, recognizing the central role that inflammatory processes can play in both oral and systemic pathophysiology.

In a later study, Ebersole et al. [22] corroborated these findings, reporting that, as periodontal disease progresses, changes in local inflammation trigger a specific host immune response, culminating in the production of serum antibodies against the bacteria involved. This evidence suggests that localized inflammation and/or infection can generate systemic manifestations. Furthermore, several acute-phase proteins, such as CRP, present significantly elevated levels in the serum of patients with periodontal disease compared with control individuals. These findings reinforce the hypothesis that chronic oral infections and inflammation associated with periodontal disease may contribute to the activation of systemic inflammatory responses.

Similarly, Ebersole et al. [23] also demonstrated that tissue destruction resulting from chronic periodontal disease intensifies the exposure of the systemic circulation to challenges that can impact the function of vascular tissues and distant organs. As a result, systemic host responses are affected by age, contributing to inflammatory and immune responses that reflect chronic oral and systemic infections.

Systemic diseases associated with periodontal disease include type 2 diabetes mellitus, atherosclerosis and cardiovascular diseases, premature birth and other gestational complications, arthritis, and certain types of cancer [27]. Among these, the bidirectional relationship between periodontal disease and type 2 diabetes mellitus is particularly well documented. Inflammatory mediators released during periodontal infection can contribute to insulin resistance and poor glycemic control, while hyperglycemia, in turn, exacerbates the inflammatory response and impairs periodontal tissue healing.

Studies in NHPs have provided important insights into this association, demonstrating how systemic metabolic changes can influence the severity and progression of periodontal disease [26,27]. These findings reinforce the relevance of NHP models in periodontal medicine and support the hypothesis that age, diet, and metabolic status are key factors modulating both local and systemic manifestations of the disease. Notably, older individuals exhibit greater alveolar bone resorption and a higher frequency of periodontal abscesses, highlighting the importance of early preventive interventions to mitigate both oral and systemic complications related to periodontal disease.

The results of the research by Jiang et al. [28] reinforce the influence of the hyperglycemic environment on the progression of periodontal disease, highlighting the intensification of the inflammatory response and the reduction in the immune defense of periodontal tissues in diabetic individuals. In the study, it was demonstrated that diabetic rhesus monkeys presented a significant increase in the levels of interleukin-17 (IL-17) and beta-defensin-3 (BD-3), indicating an exacerbated inflammatory state and an impairment of the innate immune response. These results corroborate the bidirectional relationship between diabetes mellitus and periodontal disease, suggesting that hyperglycemia not only favors tissue destruction but can also impair gingival repair mechanisms. Thus, understanding the impacts of diabetes on periodontal disease is essential for the development of more effective therapeutic strategies that consider the regulation of the inflammatory response and the preservation of periodontal tissue homeostasis in diabetic patients.

In turn, the findings of the study by Sun et al. [25] corroborate the hypothesis that periodontal disease may be associated with systemic conditions, such as metabolic syndrome (MS), through common inflammatory mechanisms. According to these authors, the higher prevalence of periodontal disease in *Macaca mulatta* with MS, associated with increased levels of C-reactive protein, suggests that chronic inflammation of the oral cavity may contribute to the activation of systemic inflammatory responses. Furthermore, the differential expression of microRNAs in subgroups of NHPs with and without the disease indicates a possible role of these molecules in the worsening of metabolic disorders in individuals with periodontal inflammation. These results reinforce the need for additional studies to elucidate the mechanisms underlying the relationship between periodontal disease and metabolic syndrome, as well as to evaluate the impact of periodontal control in the modulation of systemic inflammation associated with metabolic diseases.

The study by Raindi et al. [29] investigated the association between dental and cardiac diseases in a cohort of captive chimpanzees, but their results suggest that, despite the significant presence of supragingival plaque in chimpanzees, there was no significant correlation between dental and cardiac data. Although chimpanzees had low bleeding scores, indicating relatively good local inflammatory control, analysis of circulating markers of cardiac health revealed elevated levels of NT-proBNP in two chimpanzees that died due to cardiac disease, indicating that heart failure may be a relevant factor in deaths. The study, although it did not find a direct association between dental and cardiac diseases, reinforces the complexity of the interactions between biological systems and highlights the need for more research to understand how factors such as periodontal health may influence or be associated with systemic diseases, such as cardiac diseases, in NHPs.

Chronic periodontal disease has also been associated with an increased risk of adverse pregnancy outcomes, possibly due to systemic inflammation generated by oral infection. A study with pregnant female baboons (*Papio anubis*) analyzed the relationship between systemic inflammatory mediators and IgG antibodies against oral bacteria and the progression of induced periodontal disease. The results showed a higher incidence of low birth weight (18.1%) and reduced gestational age (9.8%) in neonates in the group with periodontal disease. In addition, the rate of spontaneous abortion, stillbirth or fetal death was higher in this group (8.7%) compared to the control (3.8%). The progression of periodontal disease in the first half of gestation was a significant risk factor, reinforcing the hypothesis that chronic oral infections can negatively influence pregnancy and highlighting the importance of their control in preventing complications [24].

Considering the growing body of evidence on the systemic repercussions of periodontal disease, it is essential to implement preventive and therapeutic measures for its management in captive non-human primates. These measures include the establishment of routine oral health assessments by trained veterinary professionals, the implementation of regular mechanical biofilm control (such as supervised tooth brushing or use of chewing devices compatible with the species), and the provision of a balanced diet that supports oral health. When indicated, periodontal treatment procedures should be incorporated into clinical protocols, alongside the use of anti-inflammatory or antimicrobial agents as appropriate. In addition, environmental enrichment and stress reduction strategies should be part of a comprehensive approach, given the known influence of systemic factors on immune response. These actions may help mitigate the local progression of periodontal disease and reduce the risk of associated systemic comorbidities in these animals.

In addition, despite the valuable contributions of the selected studies, some important limitations in the current body of research should be highlighted. The limited number of available studies in NHPs addressing the association between periodontal disease and systemic conditions restricts broader generalizations and highlights the need for further investigation. There is a lack of standardized methodologies and longitudinal studies that could more clearly establish causal relationships and better characterize disease progression. Future research should aim to explore molecular and immunological mechanisms in more depth, apply consistent diagnostic criteria, and expand the diversity of NHP species studied. These steps will be essential to advancing the understanding of the complex interactions between oral and systemic health in NHPs and to guide the development of more effective preventive and therapeutic interventions.

## 5. Conclusions

This study analyzed the relationship between periodontal disease and the emergence of systemic diseases in NHPs, revealing a consistent association between periodontal health and various systemic conditions, particularly metabolic, inflammatory and cardiovascular diseases.

Periodontal disease appears to be linked to the worsening of systemic conditions in NHPs, especially through the potential effects of chronic oral inflammation on immune and metabolic pathways. Studies suggest associations between periodontal disease and outcomes such as altered bone density, dyslipidemia, impaired immune responses, and adverse pregnancy events. These findings point to a possible role of periodontal inflammation in triggering systemic inflammatory responses that may impact general health.

Although the current evidence is primarily observational and based on a limited number of studies, it highlights the relevance of maintaining periodontal health, particularly in older individuals or those with comorbidities, as part of a broader strategy to support systemic well-being. However, we emphasize that further research—including controlled intervention studies—is needed to elucidate causal mechanisms and assess the potential therapeutic impact of periodontal treatment in NHPs.

In conclusion, this review reinforces the importance of integrating periodontal and systemic health assessments in the care of NHPs and calls for additional studies to explore the molecular and cellular mechanisms underlying this association. Such investigations may contribute to improved clinical management and welfare of captive NHPs.

## Figures and Tables

**Table 1 vetsci-12-00784-t001:** Studies relating periodontal disease and systemic diseases in non-human primates.

Year, Authors and Country	Objective	Method	Species	Associated Systemic Disease(s)
1980 Pettigrew et al. [19] United States	To determine, through radiographic examination, whether carbohydrate intolerance in diabetes mellitus was associated with radiographic evidence of alveolar bone resorption, interproximal calculus deposits, periodontal ligament space widening and radiographic carious lesions in a mixed population of diabetic and nondiabetic rhesus monkeys.	Observational and cross-sectional	*Macaca mulatta*	Diabetes mellitus
1993 Aufdemorte et al. [20] United States	To develop an animal model especially suitable for the study of osteoporosis and oral bone loss.	Experimental study	*Papio* sp.	Osteoporosis
1999 Ebersole et al. [21] United States	To address general mechanisms that could describe the association of pathological processes between periodontal disease and atherosclerosis, using a non-human primate model.	Experimental study	*Macaca fascicularis*	Atherosclerosis
2002 Ebersole et al. [22] United States	To describe results of non-human primate and human model studies to determine the effect of periodontal disease on systemic acute-phase proteins.	Experimental study	*Macaca fascicularis*	Other chronic inflammatory diseases
2008 Ebersole et al. [23] United States	To evaluate systemic inflammatory and immunological biomarkers in a cohort of *Macaca mulatta* maintained as a large family social unit, analyzing how they would be affected by age, sex and clinical oral presentation.	Observational and cross-sectional	*Macaca mulatta*	Other chronic inflammatory diseases
2014 Ebersole et al. [24] United States	To describe the relationship of systemic inflammatory mediator patterns and IgG antibodies to 20 oral bacteria in pregnant female baboons, together with clinical features of periodontal disease, as risk indicators for adverse pregnancy outcomes.	Observational and cross-sectional	*Papio anubis*	Adverse pregnancy outcomes
2014 Sun et al. [25] China	To investigate whether the non-human primate would be an appropriate animal model for the study of spontaneous periodontal disease and its association with metabolic syndrome, and whether microRNAs play roles in the co-development of metabolic disorders and periodontal disease.	Observational and cross-sectional	*Macaca mulatta*	Metabolic syndrome
2016 Lowenstine et al. [26] United States	To present a review on the comparative pathology of aging in great apes: bonobos, chimpanzees, gorillas and orangutans.	Integrative literature review	*Pan paniscus,* *Pan troglodytes* ssp., *Gorilla gorilla* ssp., *Gorilla beringei* ssp., *Pongo pygmaeus*, *Pongo abelii*	Diabetes mellitus and cardiovascular diseases
2017 Colombo et al. [27] Brazil	Identify the clinical and microbiological characteristics of naturally occurring periodontal disease in non-human primates.	Observational and cross-sectional	*Macaca mulatta*	Other chronic inflammatory diseases
2018 Jiang et al. [28] China	To develop a rhesus monkey model of diabetic periodontal disease and explore the possible mechanisms by which diabetes affects the progression of periodontal disease.	Experimental study	*Macaca mulatta*	Diabetes mellitus
2022 Raindi et al. [29] United Kingdom	To investigate the association of dental and cardiac diseases in a cohort of captive chimpanzees.	Observational and cross-sectional	*Pan troglodytes*	Cardiovascular diseases

## Data Availability

No new data were created or analyzed in this study. Data sharing is not applicable to this article.

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
