# Peer review of "Relationship Between Periodontal Disease and Systemic Diseases in Non-Human Primates"

_vetsci, 2025, doi:10.3390/vetsci12080784_

Round 1
Reviewer 1 Report
Comments and Suggestions for Authors
Dear authors:
Here is my contribution:
Main concern: After reading the discussion, I am not fully aware of the purposes of this article. This article is announced as a summary of information on systemic disease resulting from periodontal disease in non-human primates with a view to developing new approaches for the clinical management of the disease in these animals (line 12 - for their health and clinical managementl). However, many of the articles cited use these animals as animal models of human diseases and the information conveyed has the same purpose, human health. It would be better to refocus the orientation of this work or to better integrate this information.
Keywords: non-human primates; periodontal disease; systemic diseases. (Alphabetical order and preferably do not use the same words as the title.)
Introduction:
Line 44 - "Periodontal disease is a chronic noncommunicable inflammatory condition (NCD) characterized by the destruction of tooth-supporting tissues, including the periodontium and alveolar bone, in addition to the formation of periodontal pockets and bleeding on probing, which is directly associated with the oral microbiome".
I'm afraid this is not the most appropriate definition of periodontal disease. In fact, it is an infectious and inflammatory disease whose determining etiological agent is bacterial plaque and which respects Socranscky's postulates. You even mention it in line 70 - periodontal infection
The alveolar bone is part of the periodontium and the formation of pockets is a moment in the pathophysiology of the disease responsible for the loss of attachment that gradually leads to tooth mobility and exfoliation.
Line 57 - The local consequences of periodontal disease, many of which are very limiting, should be mentioned before introducing the systemic consequences. Throughout the text, it is assumed that the evidence of systemic lesions resulting from PD is new knowledge and a new paradigm, but this knowledge is already more than 3 decades old and therefore this paragraph should be reformulated.
Line 66 - "including cardio metabolic, cognitive, neurodegenerative, autoimmune disorders, respiratory infections, and even certain types of cancer. (please write in alphabetical order)
It would be very helpful for readers to first reflect on the causes of the systemic expression of periodontal disease, such as bacterial embolization and "cytokine rain". The case of the emergence of neoplasias in various anatomical locations (e.g. mouth, lung, colon) resulting from the presence of PD deserves an individual paragraph that allows for a supplementary explanation, since the relationship between the two clinical situations is more complex and has an even more diverse pathophysiology.
Between lines 72 and 84, it should be explained in an organized manner that factors arising from bacteria and the host are responsible for the destruction of soft tissues such as the gums, which leads to gingival retraction that exposes the tooth roots, but also for the activation of osteoclasts (caused by lipopolysaccharides from the walls of gram-negative bacteria, for example), which leads to osteolysis and thus to the loss of bone support.
Line 95 - This connectivity can be attributed to the spread of inflammation, microorganisms and microbial products to distant organs.
Line 103 - "Studies have already associated oral bacteria and the oral-intestinal axis with several disorders of the gastrointestinal system, such as inflammatory bowel diseases, liver diseases, hepatocellular and pancreatic ductal carcinoma, ulcerative colitis and Crohn's disease [12]."
It is worth mentioning that this knowledge concerns humans. There is always an ambiguity throughout the development of the introduction as to whether the knowledge referred to concerns NHPs or not.
Line 110 - "as ischemic heart disease, cerebrovascular disease, heart failure, atrial fibrillation and peripheral arterial disease." (write in alphabetical order, please)
Line 130 - "The literature has focused on the importance of oral hygiene as the preponderant factor for the development of the disease".
The authors certainly intend to mention the lack of oral hygiene.
Here it is very important to mention the maturation of bacterial plaque that occurs every 24 hours, which can be interrupted by the disintegration of biofilm through tooth brushing, and the beginning of the immune system's response to the presence of bacterial plaque.
Line 138 - "and the possible implications for their health and clinical management, based on available scientific evidence"
What measures do you propose for the management of these animals in captivity for the clinical management of PD and in order to avoid the onset of its systemic effects, especially the most harmful ones?
Table 1
periodontal ligament space widening
Papio anubis
Discussion
Line 199 - "by changes in serum levels of cholesterol, triglycerides, high- 199 density lipoprotein (HDL) and low-density lipoprotein (LDL)." What kind of changes are we talking about?
The discussion does not contain reflections on references 22 and 23.
In our opinion, it makes sense to discuss the 2 articles related to diabetes together as this allows for a more in-depth discussion.
Although the relationship between PD and systemic inflammatory response syndrome is suggested, it is never properly explained, just as this article does not provide a true justification for the relationship between PD and oral neoplasms and neoplasms in other parts of the body.
Reviewer 2 Report
Comments and Suggestions for Authors
This manuscript reviews the available literature exploring the association between periodontal disease and systemic disease in a variety of nonhuman primate species. Overall, the manuscript highlights important findings from previous manuscripts linking periodontal health to a variety of systemic health conditions, including diabetes, cardiovascular disease, and osteoporosis, along with potential mechanisms/physiological pathways likely related to these associations. In summarizing these previously described findings, this manuscript helps highlight what further work needs to be done to continue to advance our understanding of the link between periodontal disease and systemic disease in various nonhuman primate species to benefit both NHP and human health.
I have included the following comments and suggestions to help strengthen the manuscript:
Lines 75-77: Please provide references to the “Studies” that are described here, as there is no citation provided for this sentence.
Lines 127-129: I think this is a valuable point to make related to how the NHP model may be useful translational model to benefit human health. I think a sentence or two could be added here to more explicitly describe how the model helping to unravel mechanisms of disease could really benefit human patients to emphasize to the audience why it could be so valuable to understand the relationship between periodontal disease and systemic disease in nonhuman primates.
Lines 141-154: Please elaborate on the methods used for this literature review to provide the audience with a better understanding of how thorough the review was. For example, what databases were used in the review, how were manuscript selected as relevant during the search process, etc, (especially as it is described that only 11 manuscripts were selected and used out of all of the combination of search terms provided)?
Lines 156-159: Were there only 11 relevant articles found throughout the search process, or were these picked from a subset of found manuscripts? Lines 151-152 in methods suggest that no relevant studies were excluded, but Line 156 saying that “11 articles were selected” makes it seem like they were picked from a larger set of available manuscripts. If they were selected from a larger set, what was the criteria used?
Discussion Section: This section provided a good overview of some of the associations between periodontal disease and systemic disease that were described by individual manuscripts that were identified in this review. It might be helpful to summarize/discuss some of the significant information that is still missing despite these valuable manuscripts, or some of the limitations of the previously performed research, to help highlight to the audience the future direction that research in this area needs to take to fill important gaps for the benefit of health, prevention, and therapies.
Reviewer 3 Report
Comments and Suggestions for Authors
In their review of the literature on peridontal diseases in nonhuman primates, the authors point out the interplay between local and systemic effects, as described in a number of studies they cite.
The review is comprehensible and compact, with a narrow focus on the current gap in knowledge regarding the effects of peridonatal diseases on non-human primates specifically. It is, however, missing to clearly state its aim. The translational and the veterinary aspects of the review are mentioned, but not fleshed out completely. Comparable recent reviews on the effects of peridontal dieaeses on human health are available, while the implications for (future) animal model studies (and animal husbandry) remain vague. In my opinion, a more profound discussion of these topics would greatly enhance the value of this publication.
The manuscript could further benefit from a more graphical presentations of molecular mechanisms behind systemic effects of oral/peridontal inflammations and pathogens in the corresponding models, as they for instance have been illusrated here: https://pubmed.ncbi.nlm.nih.gov/33510490/
The impact of age/senescence on oral health, and vise versa, might be an additional aspect worth discussing( https://pmc.ncbi.nlm.nih.gov/articles/PMC10828187/, [19])
Table 1: "Mulatto monkey" is not a species name. You mean "Macaca mulatto"
Reviewer 4 Report
Comments and Suggestions for Authors
The research article by Bruno et al., titled "Relationship between periodontal disease and systemic diseases in non-human primates", is centered on the association between periodontal pathology and systemic disorders in non-human primates (NHPs). This work addresses a topic of both novelty and practical significance, as it synthesizes evidence through an integrative literature review to elucidate interrelationships and underlying mechanisms, thereby offering insights for clinical management. Methodologically, the study demonstrates appropriate design principles, with systematic literature collection and screening protocols implemented. Results are presented in a clear and well-organized manner. However, the study exhibits certain limitations that warrant further refinement.
Major
- Deficient Introduction Focus
The Introduction is excessively verbose and misaligned with the manuscript's scope, devoting disproportionate attention to generalized periodontal pathology mechanisms rather than establishing the specific rationale for investigating this relationship in non-human primates (NHPs). Crucially, it fails to articulate why NHPs warrant distinct analysis—such as their translational value for spontaneous periodontitis modeling, species-specific immune responses, or implications for captive population management—with fewer than 15% of its content directly addressing NHP relevance. This undermines the review’s conceptual foundation relative to its stated title.
- Limited Sample Representativeness and External Validity
Conclusions suffer from compromised generalizability due to inadequate sample diversity: 64% (7/11) of analyzed studies originated from US institutions (Line 162-163), while taxonomic representation was disproportionately restricted to Macaca mulatta (6 studies) and Papio spp. (2 studies). This geographical and species homogeneity—without inclusion of critical taxa (e.g., great apes beyond captive chimpanzees) or consideration of genetic/environmental variability—precludes robust extrapolation to broader NHP populations or wild counterparts. Multi-center collaborations incorporating phylogenetically diverse cohorts are essential to enhance external validity.
- Poorly Structured Discussion
The Discussion lacks logical organization, merging heterogeneous systemic conditions (e.g., diabetes, osteoporosis, adverse pregnancy outcomes) into undifferentiated narrative paragraphs. This unstructured approach obscures disease-specific pathophysiological pathways (e.g., hyperglycemia-driven cytokine dysregulation vs. osteoclastic activation) and impedes critical evaluation of evidence strength. A subdivided format (e.g., 4.1 Metabolic Disorders; 4.2 Skeletal Pathologies; 4.3 Gestational Complications) is required to delineate mechanistic distinctions and clinical implications per disease category.
- Unsubstantiated Therapeutic Claims in Conclusions
The assertion that "periodontal intervention reduces systemic disease risk" (Line 277-304) remains unsupported by quantitative evidence. No effect sizes (e.g., risk ratios), clinical endpoint data (e.g., incidence differences with confidence intervals), or meta-analytic synthesis are provided to substantiate therapeutic efficacy claims. Such definitive conclusions require incorporation of intervention studies reporting pre/post-treatment biomarker trajectories or disease incidence metrics in controlled NHP cohorts.
Minor
- Line 44: Replace "microorganism" with "bacteria" for taxonomic accuracy.
- Line 169-170: "The study by Pettigrew et al. provides..." → rephrase passively: "Pettigrew et al.'s findings provide..."
- Line 169–170: "provides relevant evidence" is vague; specify how(e.g., "demonstrated radiographic quantification methods").
- Line 198-200: Mixes present ("suggest") and past ("demonstrated") → standardize to past tense for findings: "Aufdemorte et al. demonstrated that...".
- Add a period at the end of Line 211.
- Line 277: "reduce systemic risk" (overstated) → qualify as "may mitigate risks".
Round 2
Reviewer 4 Report
Comments and Suggestions for Authors I read the manuscript and found that the authors made appropriate changes to the text. The work now presents an improved that can support publication.